# Social distancing and preventive practices of government employees in response to COVID-19 in Ethiopia

Wakgari Deressa[1]*, Alemayehu Worku[1], Workeabeba Abebe[2], Sefonias Getachew[1], Wondwossen Amogne[3]

1 Department of Preventive Medicine, School of Public Health, College of Health Sciences, Addis Ababa University, Addis Ababa, Ethiopia, 2 Department of Pediatrics and Child Health, School of Medicine, College of Health Sciences, Addis Ababa University, Addis Ababa, Ethiopia, 3 Department of Internal Medicine, School of Medicine, College of Health Sciences, Addis Ababa University, Addis Ababa, Ethiopia

* deressaw@gmail.com

**Data Availability Statement:** All relevant data are within the paper and its Supporting Information files.

## Abstract

Public health and social interventions are critical to mitigate the spread of the coronavirus disease 2019 (COVID-19) pandemic. Ethiopia has implemented a variety of public health and social measures to control the pandemic. This study aimed to assess social distancing and public health preventive practices of government employees in response to COVID-19. A cross-sectional study was conducted among 1,573 government employees selected from 46 public institutions located in Addis Ababa. Data were collected from 8th to 19th June 2020 using a paper-based self-administered questionnaire and analyzed using SPSS version 23.0. Descriptive statistics were used to summarize the data. Binary logistic regression analyses were used to identify factors associated with outcome variables (perceived effectiveness of facemask wearing to prevent coronavirus infection, and COVID-19 testing). Majority of the participants reported facemask wearing (96%), avoiding close contact with people including handshaking (94.8%), consistently followed government recommendations (95.6%), frequent handwashing (94.5%), practiced physical distancing (89.5%), avoided mass gatherings and crowded places (88.1%), restricting movement and travelling (71.8%), and stayed home (35.6%). More than 80% of the participants perceived that consistently wearing a facemask is highly effective in preventing coronavirus infection. Respondents from Oromia perceived less about the effectiveness of wearing facemask in preventing coronavirus infection (adjusted OR = 0.27, 95% CI:0.17–0.45). About 19% of the respondents reported that they had ever tested for COVID-19. Respondents between 40–49 years old (adjusted OR = 0.41, 95% CI:0.22–0.76) and 50–66 years (adjusted OR = 0.43, 95% CI:0.19–0.95) were less likely tested for coronavirus than the younger age groups. Similarly, respondents from Oromia were less likely to test for coronavirus (adjusted OR = 0.26, 95% CI:0.12–0.56) than those from national level. Participants who were sure about the availability of COVID-19 testing were more likely to test for coronavirus. About 57% of the respondents perceived that the policy measures in response to the pandemic were inadequate. The findings showed higher social distancing and preventive practices among the government employees in response to COVID-19. Rules and regulations imposed by the

**Funding:** This study was funded by Addis Ababa University (AAU) [Ref. no. RD/LT-312/2020] and partly supported by the School of Public Health. WD, AW, WAA and WA received the award from the University (RD/LT-312/2020). The funding body had no role in the study design, data collection, analysis, interpretation of data, or writing the manuscript.

**Competing interests:** The authors declare that they have no competing interests.

government should be enforced and people should properly apply wearing facemasks, frequent handwashing, social and physical distancing measures as a comprehensive package of COVID-19 prevention and control strategies.

## Introduction

The ongoing rapid spread of the coronavirus disease 2019 (COVID-19), caused by severe acute respiratory syndrome coronavirus 2 (SARS-CoV-2) [1] and first reported in China in December 2019 [2, 3], has led the World Health Organization (WHO) to declare the disease as a global pandemic on 11[th] March 2020 [4]. Since then, the virus has rapidly spread across the world and has caused over 200 million confirmed cases and more than 4.3 million deaths worldwide as of 14[th] August 2021 [5]. Almost all African countries have been hit with the pandemic with the first confirmed case reported in Egypt on 14[th] February 2020 [6]. As of 14[th] August 2021, more than 7.3 million confirmed COVID-19 cases with 183,249 deaths have been reported from Africa, with the majority of cases from South Africa, Morocco, Tunisia, Ethiopia and Egypt [5]. The virus has caused unprecedented morbidity and deaths mainly among older age people with underlying health conditions [7, 8]. However, a recent epidemiological analysis of COVID-19 cases in Ethiopia revealed that 88% of the patients were between ages 10 and 49 years, and 84% were asymptomatic, indicating that symptomatic and severe cases are lower in Ethiopia than other countries [9].

Ethiopia reported its first confirmed case of COVID-19 on 13[th] March 2020 [10]. The first case was a foreigner who tested positive by the Ethiopian Public Health Institute (EPHI). Three new secondary cases that were linked to the first case and an additional imported from Dubai were reported on 15[th] March. Since the report of the first case, updates and press statements on the situation of the pandemic in the country have been daily given to the public by the Ministry of Health (MoH) and EPHI. Most of the cases during the early phase of the pandemic were detected among people with travel history of abroad, mandatory quarantined passengers, and health screening at the points of entry to the country [11]. Within less than three months after the first case of COVID-19, the virus quickly spread to all parts of the country. By the first week of June, all regions reported COVID-19 cases, with Addis Ababa and Oromia constituting about 75% and 6% of the cases, respectively [12]. Increased number of imported cases along with increased number of secondary cases subsequently contributed to community transmission. As of 14[th] August 2021, the total number of confirmed COVID-19 cases in Ethiopia has reached 288,159 with 4,471 deaths [5], with Addis Ababa and Oromia accounting for 65% and 14% of the cases, respectively [13].

In the absence of specific therapeutics or effective immunization particularly during the early stages of a potentially pandemic outbreak such as COVID-19, public health and social measures are critical to prevent and interrupt the person-to-person transmission of the virus through respiratory droplets and close contact [14]. In order to reduce or contain the spread of SARS-COV-2, many countries have implemented a lot of public health and social measures such as isolation, quarantine, social distancing, facemask wearing and hand hygiene practices [15–17], and these measures have proved to be effective in many countries [18–21]. Mathematical models of COVID-19 transmission have predicted the impact of social distancing measures [22] and universal masking [23] on the potential reduction of the spread of coronavirus. Earlier studies also demonstrated that the global *outbreak* of severe acute respiratory syndrome (*SARS*) [24] and the pandemic influenza [25] were substantially reduced by diligent hand

hygiene practices and mask wearing. This implies that frequent handwashing with soap and water, wearing facemask, social distancing and avoiding close contacts with other people are the effective measures that can be applied by everyone to protect themselves from COVID-19.

The Government of Ethiopia has stepped up various prevention and intervention activities against COVID-19 pandemic since early February 2020. The initial containment measures used to tackle the pandemic during February and April included intense surveillance for infections, not only in incoming travelers but also screening of individuals at high risk of infection who had close contact with a confirmed case, immediate isolation of all confirmed cases, quarantine, risk communication, and a public campaign for social distancing and preventive practices. While many countries around the world have implemented drastic measures to slow down the rate of transmission of COVID-19 such as strict travel restrictions and lockdowns [26], Ethiopia has implemented a variety of less drastic essential measures in response to the spread of the virus, such as airport surveillance and suspension of flights, travel restrictions, closure of international borders, flexible working arrangements, closing schools and universities, and mandatory quarantine well ahead of many countries around the world. Religious organizations cancelled services from March 31st onwards, and sports, conferences and other mass gatherings were banned. A five-month long state of emergency was declared on 8th April 2020 [27]. At the end of May, wearing facemask both in work places and public was enforced as mandatory.

The public health and social interventions highly promoted and implemented in Ethiopia to control the rate of transmission of the SARS-CoV-2 at individual level include frequent handwashing with soap and water, social distancing, wearing facemask including home-made masks, use of alcohol-based sanitizers, staying at home when possible, covering mouth and nose while coughing and sneezing, avoiding touching the nose, mouth and eyes with hands, and refraining from risky behaviors such as travel and attending mass gatherings. Continuous investigation through laboratory testing, case detection, isolation and contact tracing have been the milestone of the control efforts throughout the country to better understand the transmission dynamics and strengthen appropriate prevention and control strategies [28].

Molecular diagnostic testing capacity in Ethiopia was very limited and initially there was no capacity for COVID-19 testing. The initial samples were transported for testing to the WHO regional reference laboratory in South Africa. The EPHI immediately launched the first COVID-19 Polymerase Chain Reaction (PCR) testing in Ethiopia during early February. Since then, the testing capacity has increased and gradually rolled out to more than 80 laboratories throughout the country. As a result, the testing has been scaled-up to over 8,000 tests per day by early August 2020. Despite all the public health measures, the number of cases has been steadily increasing in the country, but at a slower rate than the earlier catastrophic estimates [29].

Studies have shown that strong public health measures such as social distancing and other preventive behaviors have resulted in a substantial reduction in the transmission of COVID-19 [18, 22]. The impact of public health interventions and population behavioral changes that have been rolled out in Ethiopia to contain COVID-19 transmission has not been evaluated. High public compliance to proper risk reduction measures such as practicing social distancing, wearing facemask, frequent handwashing and staying home can be effectively achieved if the public understands and is persuaded of the importance of these measures in the prevention and control of COVID-19 [30]. Thus far, very limited research has reported on how individuals have practiced protective behaviors in response to COVID-19 pandemic in Ethiopia [31–33]. The aim of this study was to assess the social and public health protective measures among government employees in Addis Ababa in response to COVID-19. The results of this study are important to inform future efforts focusing on the government

employees and similar population group's readiness to comply with pandemic control measures and the development of preventive strategies and health promotion programs, given that proper practices of social distancing and preventive behaviors can play important roles in the prevention and control of COVID-19.

## Methods and materials

### Study design and setting

This cross-sectional study was conducted among government employees of 46 public institutions located in Addis Ababa. With a projected population of about 3.6 million in 2020 [34], Addis Ababa city has the highest rate of COVID-19 cases and deaths in Ethiopia, and is considered as an epicenter of COVID-19 in the country. Of the total 4,070 confirmed COVID-19 cases reported in the country as of 21st June 2020, the majority (71%) of the cases were reported from Addis Ababa. During the data collection period between 8th and 19th June 2020, the total number of confirmed new COVID-19 cases reported in the country was 2,064 including 45 deaths, of which Addis Ababa contributed 72% of the cases and 89% of the deaths. During the 12 days of data collection, the number of COVID-19 in Addis Ababa increased from 1,625 on 8th June to 2,988 on 19th June, representing an increase of 84%. The most affected sub-cities included Addis Ketema, Lideta, and Gulele, while Akaki-Kaliti sub-city had the lowest number of cases, and most of the cases were due to community transmission as of the first week of May 2020.

All the federal government offices are located in Addis Ababa city. People working in the federal government are in charge of all parts of the country and decisions made at the national level are applied at all regions. The Regional State of Oromia is the largest and most populous of the 10 states in Ethiopia, with a projected population of about 38 million in 2020 and approximately accounts for 35% of the Ethiopian population [34]. The capital city of Oromia is the national capital, Addis Ababa (a.k.a. Finfinne). All national government offices including all ministry offices, the city and sub-city administration offices, the city's sector offices and the Regional State of Oromia and its sector offices are based in Addis Ababa. Oromia is the state that surrounds Addis Ababa city, and it is one of the regions highly affected by COVID-19 [9], next to Addis Ababa. The total number of COVID-19 confirmed cases in Oromia has increased from 247 as of 21st June 2020 to 36,817 as of 14th August 2021, accounting for 6% and 14% of the national cases, respectively [13].

### Study population and sample size calculation

The study population for this study constituted all government employees working in the selected government institution at the time of the survey and willing to participate in the study. These included professionals, experts, technicians and support staff working at different hierarchies and divisions/directorates in the institution including higher and midlevel officials. The study participants were assumed to represent employees from the community in Addis Ababa and its environs involved in policy and decision-making processes. The decisions and practices made by these people would subsequently have direct or indirect influence on individuals, family and the community in response to COVID-19. Due to physical distancing restrictions, it was not possible to conduct a representative community-based face-to-face interview during this period. As a result, this study collected data using institution-based self-administered survey.

A sample size of 1,710 was calculated with a precision of 4% to estimate a 50% proportion with 95% confidence, a design effect of 2 and 30% non-response rate [35]. Purposive sampling was used to select the public institutions. The institutions were stratified into three government levels and selected from the national or federal government ministries, Addis Ababa city

administration bureaus and sub-cities, and Oromia Regional State bureaus located in Addis Ababa (S1 Appendix and S1 Fig).

## Sampling procedures

The data collectors initially contacted the respective higher official in the selected institution to explain the purpose of the survey and submit the support letter. S1 Fig shows the schematic diagram of the sampling procedures and the sample distributions. After approval of the support letter, the Human Resource Directorate of the respective institutions were contacted to obtain information on the total number of employees, number of directorates and departments in the institution with their respective number of personnel. The sample size allocated to the institution was distributed to the directorates or departments proportional to the size of their employees. Emphasis was given to equally select the participants, to proportionally distribute the number of questionnaires to the different directorates or departments in the selected institutions based on the size of their employees. A systematic random sampling technique was used to finally select the study participants in each directorate or department in the institution.

In this survey, we tried to avoid selection bias by including as many representative respondents' as possible within the shortest possible time. Since some employees were working on a shift basis due to the current situation of COVID-19 pandemic, their availability was taken into consideration while distributing the questionnaires. When the selected potential respondent was known that he/she couldn't return to the office during the first 2–3 days of the survey, replacement was made. Emphasis was also given to ensure the gender balance during the selection of the respondents and distribution of the questionnaires.

## Survey instrument and data collection

A paper-based self-administered questionnaire was used to collect data. The questionnaire was developed by the research team for the purpose of this survey, and some questions were adapted from the WHO tools used for a similar study [36]. The questionnaire had two main parts: (1) socio-demographic characteristics, and (2) social distancing and preventive practices. The tool was initially developed in English (S2 Appendix) and translated into Amharic (S3 Appendix) and Afan Oromo (S4 Appendix) by experienced personnel, and back translated into English for accuracy by independent personnel. Translators were fluent in both English as well as each local language to help ensure appropriate adaptation of the survey items.

A total of 20 trained data collectors with master's degree and previous experience were involved in data collection with 2–3 institutions per each data collector. Training and orientation on the survey including how to administer the questionnaires were conducted for the data collectors on 2nd June 2020. In addition, the Amharic and Afan Oromo versions of the questionnaires were tested on one target person by each data collector prior to the actual data collection. Few minor revisions of the instruments were made. The questionnaire included an introductory information on the cover letter to inform participants about the study and explaining the purpose of the survey, consent information to ensure voluntary participation in the study while ensuring confidentiality of data, and researchers contact information for any questions the respondent might have. Individuals who declined to participate were excluded from the survey. Participants completed the questionnaires by themselves in the local language (Afan Oromo in the Oromia Regional State Offices and Amharic otherwise). In this survey, the main role of the data collector was to ensure the selection of potential participants, to obtain informed consent, distribute the questionnaires to the respondents, and collect the completed questionnaires later on.

In this study, social distancing and preventive practices were defined as the main health protective measures that are adopted and applied by people to protect themselves and others from contracting disease such as COVID-19 and slowing down the spread of the virus [37–39]. Social distancing practices include physical distancing, staying at home if possible or when sick, working from home, avoiding mass gatherings, social events, crowded places, public transport and travelling, and avoiding close contact with people including shaking hands or hugging. Physical distancing involves the practice of maintaining at least two adult strides or two meters distance between two or more people. Preventive behaviors or hygiene practices include wearing a facemask, washing hands more frequently with water and soap, using hand sanitizer more regularly, cleaning and disinfecting surfaces including mobile phones, avoiding touching eyes, nose and mouth, and covering the mouth and nose when coughing and sneezing using a tissue paper or bent elbow. We have also intentionally included the use of "garlic, ginger and lemon' in the questionnaire to test the understanding of the study participants about the preventive measures.

## Statistical analyses

Data were entered into the Census and Survey Processing System (CSPro) software package, version 7.2 (U.S. Census Bureau and ICF Macro) and analyzed using Statistical Package for Social Sciences (SPSS) version 23 (SPSS Inc., IBM, USA). The outcome of interest was COVID-19 related protective practices (social distancing and preventive practices) taken by individuals. These variables were based on the question "Which of the following measures, if any, are you currently taking to prevent yourself against COVID-19"? Respondents were able to select from 14 possible protective health measures including staying at home, maintaining physical distancing, avoiding close contact with people including handshaking, covering mouth/nose with face/cloth mask when going outdoors, washing hands with soap and water frequently, avoiding touching eyes, nose and mouth, avoiding mass gathering, covering mouth/nose while coughing or sneezing, restricting movement, testing for coronavirus, recommending the use of facemask to people when going outdoors, and following government recommendations to combat COVID-19.

Responses to the questions were recoded as '1 = Yes' and '0 = No'. A composite index of the average of all items was created for each respondent to form total preventive measures being taken by the individual, ranging from 0 to 14, with a higher score indicating that participants demonstrate higher protective measures. The internal consistency of the items was moderate (Cronbach's alpha = 0.798). Basic descriptive statistical methods such as frequencies, percentages, means, standard deviations, and cross-tabulations were conducted to summarize the data and determine the differences between groups for selected demographic variables. Multivariable binary logistic regression models were used as measures of association between the outcome variables (perceived effectiveness of facemask wearing to prevent coronavirus infection, and ever testing for COVID-19) and the potential predictors, adjusted for potential confounders. The variance inflation factor (VIF) test was performed for age and service years as predictor variables, and no evidence of multicollinearity was detected in the regression model (VIF = 1.384). Odds ratios (ORs) and their 95% confidence intervals (CIs) for each predictor were estimated using binary logistic regression models to quantify the associations between potential predictors and outcome variables. The statistical significance level was set at $p < 0.05$.

## Ethical considerations

Ethical approval was obtained from the Institutional Review Board (IRB) of the College of Health Sciences at Addis Ababa University (protocol number: 042/20/SPH). Permission to

undertake this study was obtained from every relevant authority at all levels. Written informed consent was obtained from all study participants. All methods were performed in accordance with the relevant guidelines and regulations set out in the Declaration of Helsinki.

## Results

### Socio-demographic characteristics of study participants

In total, 1,718 eligible participants from 46 government institutions were invited to participate in the study and 1,581 participants completed the questionnaires (S1 Fig). Of these, 1,573 were valid and used for the current analysis (91.6%). About 91% of the study participants provided written informed consent, while 9% provided only verbal informed consent. The completed questionnaires per institution ranged from 18–58, with an average of 34.3. Table 1 shows the sociodemographic characteristics of the study participants. About 40% of the study participants were drawn from national institutions, 38.8% from Addis Ababa city administration institutions and 21.6% from Oromia Regional State institutions located in Addis Ababa. The majority of the respondents were in the age group of 18 and 39 years (73.3%), male (64.2%), with a bachelor's degree or above (88.3%) and lived in Addis Ababa (82.2%). The mean (±SD) year of service in the institution was 6.6 (±6.4) years. About 19% of the respondents reported that they were tested for COVID-19, 7.1% reported any chronic illness, and only 2% were quarantined due to COVID-19.

### Social distancing and preventive practices

Respondents were asked to indicate the types of protective measures they applied to prevent contracting COVID-19. Fig 1 presents the proportions of respondents who responded 'yes' to the 14 social distancing measures and other preventive practices taken in response to COVID-19. Overall, more than 9 in 10 respondents (95.9%) reported wearing facemask, 95.6% reported that they consistently followed the recommendations from the authorities to combat COVID-19, 92% reported that they recommended the wearing of facemask for healthy people out of the healthcare setting, 94.5% avoided close contact with people including handshaking, 94.1% reported frequently washing hands with water and soap, 90.8% covered mouth/nose while coughing or sneezing, 90.7% avoided touching eyes, nose and mouth, and 89.5% practiced physical distancing. The majority of the respondents also reported avoiding mass gatherings and crowded places (88.1%), disinfected surfaces (77.6%), disinfected mobile phones (76.9%), restricted movement and traveling (71.8%), ate garlic, ginger and lemon (57.9%). The lowest level of compliance in response to COVID-19 was related to staying home, which was reported by 38.5% of participants.

Table 2 shows the distribution of the responses of the respondents to the 14 social distancing and preventive measures by the government level. The majority (>90%) of the respondents reported the practice of wearing facemask, followed government recommendations, avoided close contact with people, frequently washed hands and avoided touching eyes, nose and mouth across all the three government levels. However, disinfecting surfaces (91.2%), staying home (45.7%), and restricting movement and travelling (84.1%) were more frequently reported in Oromia compared with respondents from Addis Ababa and national level. Whilst, wearing facemask (97.3%) and consistently following government recommendations (frequent handwashing with soap and water, wearing facemask, and social distancing) (97.1%) were more commonly reported among the national respondents than those from Oromia. This study also revealed that 64% of respondents from Oromia, 58.3% from national and 54% from Addis Ababa reported that they used garlic, ginger and lemon for prevention of coronavirus infection.

**Table 1. Characteristics of study participants by level of government, Addis Ababa, June 2020.**

| Characteristics | Government level, n (%) | | | Total, n (%) |
|---|---|---|---|---|
| | **National** | **Oromia** | **Addis Ababa** | |
| **Gender** | | | | |
| Male | 405 (64.9) | 228 (67.3) | 350 (57.4) | 983 (62.5) |
| Female | 206 (33.0) | 99 (29.2) | 244 (40) | 549 (34.9) |
| Unknown[a] | 13 (2.1) | 12 (3.5) | 16 (2.6) | 41 (2.6) |
| **Age group (years)** | | | | |
| 18–29 | 174 (27.9) | 48 (14.2) | 175 (28.7) | 397 (25.2) |
| 30–39 | 253 (40.5) | 145 (42.8) | 258 (42.3) | 656 (41.7) |
| 40–49 | 90 (14.4) | 77 (22.7) | 94 (15.4) | 261 (16.6) |
| ≥50 | 55 (8.8) | 35 (10.3) | 32 (5.2) | 122 (7.8) |
| Unknown | 52 (8.3) | 34 (10.0) | 51 (8.4) | 137 (8.7) |
| **Level of education** | | | | |
| ≤12th grade | 17 (2.7) | 7 (2.1) | 23 (3.8) | 47 (3.0) |
| Diploma[b] | 53 (8.5) | 14 (4.1) | 65 (10.7) | 132 (8.4) |
| Bachelor's degree | 318 (51.0) | 180 (53.1) | 391 (64.1) | 889 (56.5) |
| Master's degree or above | 226 (36.2) | 121 (35.7) | 126 (20.7) | 473 (30.1) |
| Unknown | 10 (1.6) | 17 (5.0) | 5 (0.8) | 32 (2.0) |
| **Year of experience in the institution** | | | | |
| <5 | 352 (56.4) | 82 (24.2) | 228 (53.8) | 762 (48.4) |
| 5–9 | 160 (25.6) | 79 (23.3) | 167 (27.4) | 406 (25.8) |
| 10–14 | 44 (7.1) | 76 (22.4) | 61 (10.0) | 181 (11.5) |
| ≥15 | 51 (8.2) | 78 (23.0) | 39 (6.4) | 168 (10.7) |
| Unknown | 17 (2.7) | 24 (7.1) | 15 (2.5) | 56 (3.6) |
| **Household size** | | | | |
| 1–3 | 267 (42.8) | 89 (26.3) | 227 (37.2) | 583 (37.1) |
| 4–5 | 231 (37.0) | 142 (41.9) | 249 (40.8) | 622 (39.5) |
| 6–7 | 76 (12.2) | 65 (19.2) | 85 (13.9) | 226 (14.4) |
| ≥8 | 24 (3.8) | 17 (5.0) | 34 (5.6) | 75 (4.8) |
| Unknown | 26 (4.2) | 26 (7.7) | 15 (2.5) | 67 (4.3) |
| **Area of residence** | | | | |
| Addis Ababa city | 580 (92.9) | 129 (38.1) | 584 (95.7) | 1293 (82.2) |
| Out of Addis Ababa | 37 (5.9) | 147 (43.3) | 25 (4.1) | 209 (13.3) |
| Unknown | 7 (1.1) | 63 (18.6) | 1 (0.2) | 71 (4.5) |
| **Tested for COVID-19** | | | | |
| Yes | 156 (25.0) | 26 (7.7) | 118 (19.3) | 300 (19.1) |
| No | 448 (71.8) | 251 (74.0) | 476 (78.0) | 1175 (74.7) |
| Unknown | 20 (3.2) | 62 (18.3) | 16 (2.6) | 98 (6.2) |
| **Reported any chronic illness** | | | | |
| Yes | 35 (5.6) | 33 (9.7) | 44 (7.2) | 112 (7.1) |
| No | 483 (77.4) | 232 (68.4) | 441 (72.3) | 1156 (73.5) |
| Don't know or unknown | 106 (17.0) | 74 (21.8) | 125 (20.5) | 305 (19.4) |
| **Quarantined due to COVID-19** | | | | |
| Yes | 15 (2.4) | 8 (2.4) | 8 (1.3) | 31 (2.0) |
| No | 605 (97.0) | 318 (93.8) | 592 (97.0) | 1515 (96.3) |
| Unknown | 4 (0.6) | 13 (3.8) | 10 (1.6) | 27 (1.7) |
| **Total, n (%)** | **624 (100)** | **339 (100)** | **610 (100)** | **1,573 (100)** |

[a]Non-response.

[b]12th grade complete and one or more years of training.

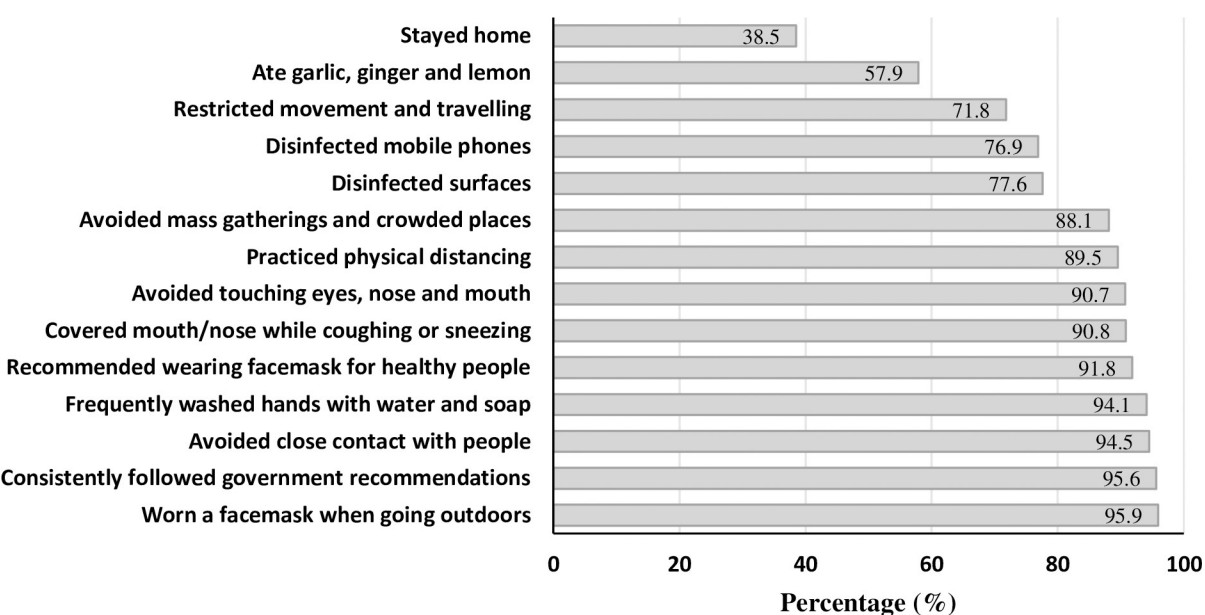

**Fig 1. The proportion of respondents who responded to the 14 social and preventive practices in response to COVID-19, Addis Ababa, June 2020.**

The responses of the 14 social distancing and preventive practices in response to COVID-19 were added to produce the score of the overall reported practice. Summing the responses across the total measures for each individual revealed a mean sample score of 11±2.3 measures taken and a median of 12 measures with an interquartile range (IQR) of 2, on a scale of 14. About 24% and 20% of the respondents scored 13 and 14 responses on the social and preventive measures, respectively, while 10.6% responded 8 or less types of measures taken in

**Table 2. Distribution of respondents reported the social and protective measures in response to COVID-19 by government level (n = 14 items), Addis Ababa, June 2020.**

| Reported social and protective practices | Government level, % | | |
|---|---|---|---|
| | **National** | **Oromia** | **Addis Ababa** |
| Worn a facemask when going outdoors | 97.3 | 94.1 | 95.6 |
| Consistently followed government recommendations | 97.1 | 93.8 | 95.1 |
| Avoided close contact with people and hand shaking | 95.7 | 92.0 | 94.8 |
| Frequently washed hands with water and soap | 95.4 | 91.7 | 94.1 |
| Recommended wearing facemask for healthy people | 93.9 | 85.8 | 93.0 |
| Covered mouth/nose while coughing or sneezing | 93.6 | 90.3 | 88.8 |
| Avoided touching eyes, nose and mouth | 92.0 | 90.0 | 89.7 |
| Practiced physical distancing | 90.9 | 89.7 | 88.0 |
| Avoided mass gatherings and crowded places | 91.8 | 83.5 | 86.9 |
| Disinfected surfaces | 75.5 | 91.2 | 72.1 |
| Disinfected mobile phones | 81.1 | 79.6 | 71.1 |
| Restricted movement and travelling | 71.3 | 84.1 | 65.4 |
| Ate garlic, ginger and lemon | 58.3 | 64.0 | 53.9 |
| Stayed home | 37.3 | 45.7 | 35.6 |
| **Total, n** | **624** | **339** | **610** |

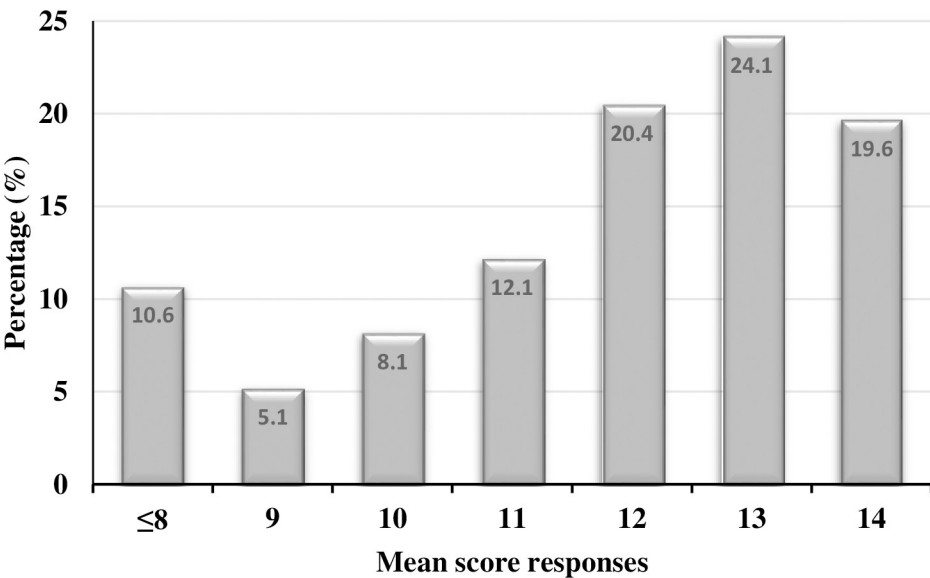

**Fig 2. Mean score responses of respondents from 14 social and preventive practices of COVID-19, Addis Ababa, June 2020.**

response to COVID-19 (Fig 2). Overall, 64% of the respondents gave 12 or more responses out of the 14 protective measures.

## Perceived effectiveness of mask wearing and associated factors

About 80% of the participants agreed or strongly agreed that consistently wearing a facemask is highly effective in preventing the transmission of coronavirus, while 10.3% were unsure. Only 9.3% of the respondents disagreed or strongly disagreed about the effectiveness of facemask in the prevention of coronavirus infection. Nearly 21% of the respondents in Oromia neither agreed nor disagreed about the effectiveness of facemasks in the prevention of coronavirus infection.

Table 3 shows the findings of the bivariate and multivariable logistic regression analyses of predictors associated with the respondent's perceived effectiveness of consistently wearing facemask in preventing the infection due to coronavirus. Ten predictors were initially assessed by bivariate analyses and retained in the multivariate logistic regression model. Among the predictors assessed by the bivariate logistic regression analyses, respondents who served in the institution for more than 15 years (OR = 0.68, 95% CI:0.45–1.03) and those from Oromia (OR = 0.31, 95% CI:0.22–0.43) were less likely to perceive the effectiveness of facemask, whilst, study participants who resided in Addis Ababa (OR = 2.02, 95% CI:1.44–2.83) and who were tested for coronavirus (OR = 1.43, 95% CI:1.00–2.04) were more likely to perceive the effectiveness of facemask to prevent coronavirus infection. In the multivariable logistic regression, participants from Oromia, compared to the national respondents, reported statistically significantly lower odds (adjusted OR = 0.27, 95% CI:0.17–0.45) of perceived effectiveness of facemask in preventing coronavirus infection. However, the other predictor variables were not statistically significant in the multivariate analyses.

## COVID-19 testing and associated factors

About 19% (n = 300) of the respondents reported that they had ever tested for coronavirus infection, with 25% of respondents at national level, 19.3% from Addis Ababa and 7.7% from

**Table 3. Factors associated with perceived effectiveness of facemask in preventing coronavirus infection using bivariate and multivariable logistic regression analyses, Addis Ababa, June 2020.**

| Predictor | Total | Highly effective (%) | Crude OR (95% CI)[a] | P-value | Adjusted OR (95% CI) | P-value |
|---|---|---|---|---|---|---|
| **Gender** | | | | | | |
| Male | 969 | 80.7 | 0.85 (0.65–1.12) | 0.251 | 0.75 (0.52–1.08) | 0.117 |
| Female | 544 | 83.1 | 1[b] | | 1 | |
| **Age group (year)** | | | | | | |
| 18–29 | 395 | 81.3 | 1 | | 1 | |
| 30–39 | 649 | 82.9 | 1.12 (0.81–1.55) | 0.503 | 1.24 (0.82–1.88) | 0.307 |
| 40–49 | 256 | 78.9 | 0.86 (0.58–1.28) | 0.460 | 0.96 (0.56–1.63) | 0.868 |
| 50–66 | 119 | 83.2 | 1.14 (0.66–1.96) | 0.634 | 1.40 (0.68–2.88) | 0.365 |
| **Education** | | | | | | |
| ≤12th grade | 47 | 89.4 | 1 | | 1 | |
| Diploma | 131 | 79.4 | 0.46 (0.17–1.27) | 0.134 | 0.41 (0.11–1.58) | 0.196 |
| Bachelor's degree | 879 | 80.0 | 0.48 (0.19–1.22) | 0.122 | 0.63 (0.18–2.19) | 0.463 |
| ≥Master's degree | 465 | 84.3 | 0.64 (0.25–1.66) | 0.639 | 0.91 (0.25–3.28) | 0.885 |
| **Experience (year)** | | | | | | |
| <5 | 754 | 83.3 | 1 | | 1 | |
| 5–9 | 401 | 81.0 | 0.86 (0.63–1.18) | 0.340 | 0.83 (0.57–1.22) | 0.351 |
| 10–14 | 180 | 81.7 | 0.89 (0.59–1.37) | 0.603 | 1.25 (0.72–2.18) | 0.436 |
| ≥15 | 162 | 77.2 | 0.68 (0.45–1.03) | 0.065 | 0.85(0.46–1.57) | 0.613 |
| **Household size** | | | | | | |
| 1–3 | 577 | 81.6 | 1 | | 1 | |
| 4–5 | 616 | 82.8 | 1.08 (0.81–1.46) | 0.600 | 1.30 (0.91–1.87) | 0.149 |
| 6–7 | 219 | 80.8 | 0.95 (0.64–1.41) | 0.794 | 1.06 (0.65–1.72) | 0.820 |
| ≥8 | 74 | 79.7 | 0.89 (0.48–1.62) | 0.693 | 1.19 (0.57–2.49) | 0.647 |
| **Government level** | | | | | | |
| National | 621 | 87.0 | 1 | | 1 | |
| Oromia | 328 | 67.4 | 0.31 (0.22–0.43) | **<0.001** | 0.27 (0.17–0.45) | **<0.001** |
| Addis Ababa | 603 | 83.6 | 0.76 (0.56–1.05) | 0.096 | 0.80 (0.56–1.15) | 0.234 |
| **Residence** | | | | | | |
| Addis Ababa | 1278 | 83.7 | 2.02 (1.44–2.83) | **<0.001** | 1.04 (0.63–1.71) | 0.885 |
| Out of Addis Ababa | 206 | 71.8 | 1 | | 1 | |
| **Tested for COVID-19** | | | | | | |
| Yes | 297 | 85.9 | 1.43 (1.00–2.04) | 0.052 | 1.47 (0.95–2.27) | 0.082 |
| No | 1162 | 81.0 | 1 | | 1 | |
| **Reported any chronic illness** | | | | | | |
| Yes | 110 | 81.0 | 1.03 (0.62–1.70) | 0.925 | 1.18 (0.64–2.17) | 0.597 |
| No or didn't know | 1375 | 81.5 | 1 | | 1 | |
| **Quarantined due to COVID-19** | | | | | | |
| Yes | 31 | 77.4 | 0.77 (0.33–1.81) | 0.555 | 0.57 (0.17–1.92) | 0.365 |
| No | 1499 | 81.6 | 1 | | 1 | |

[a]OR = Odds Ratio, CI = Confidence Interval.

[b]Reference.

Oromia. On the certainty of getting a COVID-19 test if needed, 11% of the respondents were completely sure, 18.1% were very sure, and 26% were somewhat sure. Table 4 shows the results of the bivariate and multivariable logistic regression analyses conducted to explore factors associated with COVID-19 testing. In the bivariate analyses gender, age, year of experience,

**Table 4. Factors associated with coronavirus testing in the study population using bivariate and multivariable logistic regression analyses, Addis Ababa, June 2020.**

| Predictor | Total | Tested (%) | Crude OR (95% CI)[a] | P-value | Adjusted OR (95% CI) | P-value |
|---|---|---|---|---|---|---|
| **Gender** | | | | | | |
| Male | 924 | 18.7 | 0.76 (0.58–0.99) | **0.039** | (0.68–1.47) | 0.998 |
| Female | 515 | 23.3 | 1[b] | | 1 | |
| **Age group (year)** | | | | | | |
| 18–29 | 387 | 28.2 | 1 | | 1 | |
| 30–39 | 612 | 17.6 | 0.45 (0.40–0.74) | **<0.001** | 0.70 (0.46–1.06) | 0.095 |
| 40–49 | 235 | 14.0 | 0.42 (0.27–0.64) | **<0.001** | 0.41 (0.22–0.76) | **0.005** |
| 50–66 | 118 | 16.1 | 0.49 (0.29–0.84) | **<0.001** | 0.43 (0.19–0.95) | **0.038** |
| **Education** | | | | | | |
| ≤12th grade | 42 | 23.8 | 1 | | 1 | |
| Diploma | 122 | 36.1 | 1.81 (0.81–4.02) | 0.148 | 2.12 (0.66–6.80) | 0.208 |
| Bachelor's degree | 841 | 20.1 | 0.81 (0.39–1.67) | 0.560 | 0.61 (0.21–1.84) | 0.382 |
| ≥Master's degree | 448 | 16.3 | 0.21 (0.29–1.32) | 0.218 | 0.56 (0.18–1.74) | 0.318 |
| **Experience (year)** | | | | | | |
| <5 | 738 | 21.7 | 1 | | 1 | |
| 5–9 | 382 | 21.5 | 0.99 (0.73–1.33) | 0.934 | 1.37 (0.91–2.05) | 0.132 |
| 10–14 | 165 | 13.9 | 0.59 (0.36–0.94) | **0.027** | 0.95 (0.48–1.86) | 0.870 |
| ≥15 | 146 | 19.5 | 0.82 (0.52–1.29) | 0.390 | 1.71 (0.84–3.48) | 0.136 |
| **Household size** | | | | | | |
| 1–3 | 552 | 22.1 | 1 | | 1 | |
| 4–5 | 586 | 19.1 | 0.83 (0.63–1.11) | 0.213 | 0.78 (0.53–1.14) | 0.201 |
| 6–7 | 209 | 19.1 | 0.83 (0.56–1.24) | 0.373 | 0.99 (0.59–1.67) | 0.979 |
| ≥8 | 72 | 18.1 | 0.78 (0.41–1.46) | 0.434 | 0.91 (0.39–2.13) | 0.827 |
| **Government level** | | | | | | |
| National | 604 | 25.8 | 1 | | 1 | |
| Oromia | 277 | 9.4 | 0.30 (0.19–0.43) | **<0.001** | 0.26 (0.12–0.56) | **0.001** |
| Addis Ababa | 594 | 19.9 | 0.71 (0.54–0.93) | **0.014** | 0.70 (0.49–0.99) | **0.044** |
| **Residence** | | | | | | |
| Addis Ababa | 1236 | 21.5 | 1.88 (1.19–2.98) | **0.007** | 0.88 (0.45–1.69) | 0.693 |
| Out of Addis Ababa | 181 | 12.7 | 1 | | 1 | |
| **Reported any chronic illness** | | | | | | |
| Yes | 106 | 12.3 | 0.53 (0.29–0.97) | **0.038** | 0.75 (0.38–1.46) | 0.391 |
| No or didn't know | 1313 | 20.8 | 1 | | 1 | |
| **Quarantined due to COVID-19** | | | | | | |
| Yes | 25 | 40.0 | 2.66 (1.18–5.97) | **0.018** | 1.10 (0.31–3.93) | 0.888 |
| No | 1435 | 20.1 | 1 | | 1 | |
| **Certainty to get COVID-19 test** | | | | | | |
| Not sure or didn't know | 440 | 13.6 | 1 | | 1 | |
| Only a little sure | 166 | 21.1 | 1.70 (1.07–2.69) | **0.021** | 1.75 (0.95–3.21) | 0.073 |
| Somewhat sure | 387 | 24.3 | 2.03 (1.42–2.91) | **<0.001** | 1.94 (1.22–3.08) | **0.005** |
| Very sure | 272 | 22.4 | 1.83 (1.24–2.72) | **0.003** | 1.81 (1.11–2.96) | **0.018** |
| Completely sure | 174 | 23.0 | 1.90 (1.21–2.95) | **0.005** | 1.82 (1.04–3.17) | **0.036** |

[a]OR = Odds Ratio, CI = Confidence Interval.

[b]Reference.

level of government, area of residence, reported chronic illness, being quarantined and certainty of getting COVID-19 testing if needed were significantly associated with COVID-19 testing. In the multivariable logistic regression analyses, the age groups 40–49 (adjusted OR = 0.41, 95% CI:0.22–0.76) and 50–66 years old (adjusted OR = 0.43, 95% CI:0.19–0.95) were less likely to test for coronavirus than the younger age groups. Similarly, respondents from Oromia were less likely to test for coronavirus (adjusted OR = 0.26, 95% CI:0.12–0.56) than those from national level. Furthermore, respondents who were sure about the certainty of getting COVID-19 testing if needed were more likely to report COVID-19 test. Gender, educational status, year of experience, household size, residence, reported chronic illness and being quarantined did not appear statistically significant in the multivariable logistic regression model to predict the odds of testing for COVID-19.

## Perceived adequacy of policy responses

Table 5 shows the perceptions of the respondents towards the policy decisions made by the government to contain the spread of COVID-19 pandemic in Ethiopia. Just under the third (31.3%) of the respondents strongly agreed that the policy responses that the government had taken to contain the spread of coronavirus were fair and reasonable, and 38.5% agreed with the policy responses. However, 22.8% of the respondents in Oromia disagreed about the fairness and reasonability of policy responses taken by the government. Over half (57.1%) of the study participants perceived that the policy measures taken by the government to contain the spread of coronavirus are inadequate (37.7%) or very inadequate (19.4%). More respondents from Oromia (63.7%) as compared with 59.3% in Addis Ababa and just about half (51.1%) at national level perceived that the current policy measures taken by the government to contain the spread of coronavirus transmission were inadequate.

**Table 5. Perception of respondents about policy responses to contain the spread of coronavirus in Ethiopia, Addis Ababa, June 2020.**

| Question | Government level, % | | | Total, % |
|---|---|---|---|---|
| | National | Oromia | Addis Ababa | |
| **Policy responses that have been made by the government to contain the spread of coronavirus are fair and reasonable** | | | | |
| Strongly agree | 32.1 | 32.7 | 29.8 | 31.3 |
| Agree | 42.1 | 26.8 | 41.3 | 38.5 |
| Neither agree nor disagree | 11.5 | 13.9 | 9.2 | 11.1 |
| Disagree | 6.6 | 8.6 | 10.8 | 8.6 |
| Strongly disagree | 6.6 | 14.2 | 7.7 | 8.6 |
| Unknown[a] | 1.1 | 3.8 | 1.1 | 1.7 |
| **What do you think about the adequacy of current measures by the government to contain the spread of coronavirus in Ethiopia?** | | | | |
| Very adequate | 4.5 | 4.7 | 5.4 | 4.9 |
| Adequate | 21.5 | 17.1 | 12.0 | 16.8 |
| Neither adequate nor inadequate | 21.6 | 10.6 | 20.8 | 18.9 |
| Inadequate | 34.3 | 43.1 | 38.2 | 37.7 |
| Very inadequate | 16.8 | 20.6 | 21.3 | 19.4 |
| Unknown | 1.3 | 3.8 | 2.3 | 2.2 |
| **Total, n** | **624** | **339** | **610** | **1,573** |

[a]Non-response

## Discussion

The findings of the current study revealed high level of reported practices of COVID-19 social and protective measures, particularly with regard to mask wearing in public (95.9%), avoiding close contact with people and handshaking (94.5%), frequent handwashing with water and soap (94.1%), maintaining proper physical distancing (89.5%), avoiding crowds and mass gatherings (88%), and movement restriction (72%). A study conducted during the early phase of the pandemic in Ethiopia reported a lower level of protective behaviors against the COVID-19 infection such as washing hands frequently (77%), avoiding shaking hands (54%) and not going to crowded places (33%) [31]. Another study conducted during April 2020 within high-risk population groups in Addis Ababa reported that only 49% of respondents practiced preventive measures towards COVID-19 [40]. It has been observed that staying at home, maintaining physical distancing, avoiding public transport and not going to public places is particularly difficult for many government employees, resulting in less adoption of these social distancing measures.

A study conducted during the early stage of the COVID-19 epidemic in China identified that respondents adopted important protective behaviors, and nearly all study participants (98%) wore masks when going out in public during the study period [41]. In Hong Kong, individual behaviors in the population changed in response to the threat of COVID-19, and 85% of respondents reported avoiding crowded places and 99% reported wearing facemasks in public [18]. Until effective treatments and vaccines are available, behavioral interventions such as social distancing and preventive practices are the most recommended tools to prevent and control new pandemics such as COVID-19 [42]. Studies found that the non-pharmaceutical interventions (including border restrictions, quarantine, isolation, physical distancing, and changes in population behavior) were substantially associated with reduced transmission of COVID-19 [18]. Maintaining and sustaining high levels of actual protective practices, particularly wearing mask at public, is critically important and concerted efforts should be made by the government, media, healthcare professionals, local organizations, the community and individuals to combat COVID-19 focusing on preventive health behaviors.

This study revealed higher levels of social and preventive practices in response to COVID-19. One potential reason for this higher degree of practices in the current study was due to higher level of perceived risk of COVID-19 associated with widespread information about the pandemic provided to the public. A number of studies reported higher levels of perceived risk towards COVID-19 associated with increased preventive practices, such as in Turkey [43], Hong Kong [44] and the US [45]. In contrast, a study conducted towards the end of April 2020 in northwest Ethiopia revealed a lower level (23%) of perceived risk about COVID-19 [46], which might be explained by the differences in measurements, study population or low levels of risk communication activities in the area. The finding of the current study is encouraging and may highlight the effectiveness of risk communication interventions extensively implemented in Addis Ababa during the early phase of the pandemic.

In the current study, respondents from Oromia reported a lower proportion of implementing the COVID-19 preventive practices than those from national or Addis Ababa employees. One possible explanation could be that about half of the participants from Oromia resided out of Addis Ababa where the initial efforts of the pandemic prevention focused on Addis Ababa. The national and Addis Ababa offices implemented the prevention practices earlier than Oromia offices, leading to the higher proportion of the preventive practices in the former offices. Since Addis Ababa was the epicenter of the pandemic during the early phases of the pandemic, the national and Addis Ababa offices might have received higher attention than Oromia offices. However, the uptake of preventive measures by people might be lower at the latter stages of the pandemic.

According to the current study, about 80% of the respondents perceived that consistently wearing facemask is highly effective in preventing the spread of coronavirus, and 92% supported its use by healthy people in public. At the time of data collection for this study, discussions were underway whether facemasks should be used in public by healthy people out of the healthcare setting. The WHO earlier in April advised not to use facemasks in the community setting by healthy individuals without respiratory symptoms [16], but later recommended universal masking in June [17] when asymptomatic and pre-symptomatic infectiousness of SARS-CoV-2 was established [47, 48]. Until recently, masks have mainly been worn by individuals in the general community who have certain respiratory symptoms and by those who feel particularly susceptible to infection and want to protect themselves [18].

Studies from China [21] and the US [49] have shown wearing masks is effective in reducing the risk of infection and mitigating the spread of COVID-19, particularly when combined with other preventive measures such as physical distancing and frequent handwashing. A recent systematic review and meta-analysis study funded by the WHO demonstrated the effectiveness of physical distancing of 1m or more and the use of facemasks in public and health-care settings in the prevention of coronavirus transmission, where both interventions reduced the risk of infection of coronavirus by more than 80% [50]. A case-control study from Thailand found that mask wearing, frequent handwashing and social distancing of ≥1m were independently and significantly associated with reduced risk of SARS-CoV-2 infection among the general public [51]. Previous studies also showed that the use of facemasks among the general population significantly reduced total infections and the number of deaths, and mask wearing is considered as one of the most effective public health measures in mitigating transmission of SARS-CoV-2 [52]. Studies from various countries showed the effectiveness of the national application of social distancing measures in reducing the spread of SARS-CoV-2 [53–55]. These studies provide the most reliable evidence on the effectiveness of the use of social distancing measures at community level to mitigate the spread of COVID-19.

The current study also assessed factors affecting the perceived effectiveness of mask wearing against the prevention of coronavirus infection. It was found that respondents from Oromia were less likely to report the effectiveness of wearing facemasks to prevent the transmission of SARS-CoV-2 than those from national or Addis Ababa levels, which might be associated with inadequate awareness or knowledge about the protective benefits of wearing facemask against the infection of SARS-COV-2. The more people became aware of the risk of COVID-19 to themselves, the more likely they begin practicing protective behaviors like mask wearing, handwashing and social distancing. In Ethiopia, mask wearing in public was not a common practice before the occurrence of COVID-19 pandemic. Even after the onset of the pandemic, mask wearing practice in public was very minimal, but it increased immediately when the mandatory policy of facemask wearing for all people in public and working places was enforced at the end of May 2020.

The present study also showed that respondents aged between 18 and 29 years were more likely to be tested for COVID-19 compared to the older respondents, while study participants from Oromia were less likely to be tested compared with respondents from national levels, which could be associated with lower awareness about access to the COVID-19 testing. The study also revealed that respondents who were certain about the availability of COVID-19 testing were more likely to be tested, which indicates that if people are more knowledgeable about the availability of testing centers, they are more likely to be tested. Testing can help people determine if they are infected with SARS-CoV-2 regardless of whether they have symptoms and particularly to self-isolate themselves at the time of most infectious period to minimize the risk of spreading the infection to others and to inform public health decisions [56].

Behavioral changes are currently one of the main tools to fight against COVID-19 [18–21]. These changes include practicing social and physical distancing, frequent handwashing, using hand sanitizers, wearing facemasks and testing for COVID-19 [15–18]. Nonetheless, these behavioral measures are effective if they are widely accepted and applied by the public. To have these measures widely understood and implemented by the community, the government of Ethiopia needs to heighten risk communication activities to educate the public about the significance of frequent handwashing, wearing facemasks, and social distancing in containing the transmission of SARS-COV-2.

A significant proportion (58%) of respondents in the current study reported that they used garlic, ginger and lemon to protect themselves against SARS-COV-2 infection, which indicates unconfirmed practices or misconceptions. Although these home remedies are important ingredients of our daily food and may have some medicinal properties, it is a great misconception to believe and use them against COVID-19 since they have not been tested against SARS-COV-2. Studies have found no evidence that the use of herbal remedies such as garlic and ginger is effective against infection from coronavirus or cure from COVID-19 [57, 58]. The WHO has also confirmed that there is no evidence that eating garlic or ginger has protected people from SARS-COV-2 infection [59]. Current evidence shows that using ginger or garlic or combining them with other ingredients, such as lemon, or drinking hot ginger tea does not prevent or cure COVID-19.

At the time of this study, the spread of COVID-19 in Addis Ababa city cumulatively increased from 1,625 on 8th June to 2,988 on 19th June 2020, with an average of 114 per day. Despite efforts to contain and mitigate the transmission of coronavirus in the city, the virus has continued to spread to all parts of the city at an alarming rate and more cases from the community have continued to emerge on a daily basis. COVID-19 appeared to quickly spread in Ethiopia through the movement and frequent contact between people. Physical distancing has remained a major challenge due to overcrowding and, people are confronted with the logistical and communication problems particularly due to the shortage of means of transportation, although the state of emergency mandated all vehicles to reduce the number of passengers by half. Staying home approaches were particularly challenged in the context of poverty in the city where many residents lack adequate shelter, sanitation, and economic means for livelihood. Although staying home and physical distancing slows the transmission of SARS-COV-2, they result in heavy toll particularly on the informal economic and casual labor sector due to search of income for the day-to-day livelihood [26].

Ethiopia declared a five-month state of emergency in April 2020 to mitigate the spread of COVID-19 pandemic [27]. Mandatory facemask wearing at banks, marketplaces, transport depots, in public transit, shops, pharmacies, places where public services are provided or any other public space of mass gatherings was mandated in the state of emergency. In addition, the government also made mandatory facemask wearing for all people outside of their homes or offices on 27th May. As a result, the practice of social distancing measures and preventive behaviors such as mask wearing in public were significantly improved until the state of emergency was lifted on 11th September 2020. Unfortunately, this was followed by the roll back of the already adopted social distancing measures and preventive practices by the public.

We might argue that the perceptions or preventive practices during the early period would be higher when people's tensions, worry, fear and concerns were high at the beginning. Later on, people might be reluctant and the perceptions and practices might be lower when the government relaxed the restrictions as compared to the earlier times. Consequently, the EPHI adopted a directive on 5th October 2020, which enforced mandatory wearing of facemasks in public and working places, maintaining physical distancing of at least 2m apart from other people; regular handwashing with soap or alcoholic-based sanitizers; and prohibited any

organization to provide service to any person who is not wearing a facemask [60]. However, these measures have not been well enforced and the public has become reluctant regarding the social distancing and preventive practices of COVID-19.

As the practice of social distancing involves staying home and away from others as much as possible to help prevent spread of COVID-19, Ethiopia mainly promoted physical distancing which involves the need to stay at least 2m from others, complemented by wearing facemasks. Studies have shown the effectiveness of social distancing and mandatory facemask in public in mitigating the spread of COVID-19 in many countries, and both interventions and the simultaneous implementation of other preventive measures have been identified as the strategic priorities for containing COVID-19 [61]. Although the current vaccines are effective against the consequences of COVID-19, most low-income and middle-income countries face difficulties in accessing vaccines to their populations [62]. Thus, the intensive implementation of the proven social and preventives practices such as wearing facemask, handwashing, maintaining physical distancing, and avoiding mass gatherings and crowded places is still very critical. In addition, many new variants of SARS-CoV-2 with increased transmissibility and disease severity have been recently emerged in many countries around the world, which can lead to significant clinical, therapeutic and public health impacts [63–65].

## Limitations

This study had some limitations including selection bias that deserve explanations. First, the study only included government employees working in Addis Ababa city, and it failed to include unemployed people or other individuals working in non-governmental or private institutions, leading to concerns about the representativeness of the sample. There might be differences in adapting protective health measures between employed and unemployed people as well as between employees of governmental and non-governmental institutions. Second, the data presented in this study are based on retrospective self-reports of respondents without verification, thus the results might be subjected to social desirability and recall biases. The respondents might over estimate their practices.

## Conclusions

This study has generated valuable information about public health and social measures against COVID-19 among government employees. The findings showed higher social distancing and preventive practices in response to COVID-19. In the current pandemic scenario, people should follow the governments' instructions and properly apply social distancing measures, wearing facemasks, and washing hands frequently with water and soap. Rules and regulations imposed by the government should be properly enforced in order to control the pandemic. The findings have significant implications in highlighting the importance of promoting compliance with recommended protective health behaviors to effectively control the ongoing COVID-19 pandemic. The results of this study can also be used as a baseline data to the government, other stakeholders involved in the prevention and control of COVID-19 and researchers for other larger studies to identify factors significantly associated with preventive health measures in order to implement better intervention approaches.

## Supporting information

**S1 Appendix. List of institutions/organizations included in the survey with collected samples, June 2020.**
(PDF)

**S2 Appendix. English version of self-administered questionnaire developed for the survey, June 2020.**
(PDF)

**S3 Appendix. Amharic version of self-administered questionnaire used for the survey, June 2020.**
(PDF)

**S4 Appendix. Afan Oromo version of self-administered questionnaire used for the survey, June 2020.**
(PDF)

**S1 Fig. Schematic design of the sampling methods and sample distribution, June 2020.**
(TIF)

**S1 Dataset.**
(SAV)

## Acknowledgments

The authors are grateful to the research staff at the College of Health Sciences. The authors would also like to thank the data collectors and study participants for their time and contributing to the research.

## Author Contributions

**Conceptualization:** Wakgari Deressa, Alemayehu Worku, Workeabeba Abebe, Wondwossen Amogne.

**Data curation:** Wakgari Deressa, Alemayehu Worku.

**Formal analysis:** Wakgari Deressa, Alemayehu Worku.

**Funding acquisition:** Wakgari Deressa, Workeabeba Abebe, Wondwossen Amogne.

**Investigation:** Wakgari Deressa.

**Methodology:** Wakgari Deressa, Alemayehu Worku, Workeabeba Abebe, Sefonias Getachew, Wondwossen Amogne.

**Project administration:** Wakgari Deressa.

**Resources:** Wakgari Deressa.

**Supervision:** Wakgari Deressa, Alemayehu Worku, Sefonias Getachew.

**Writing – original draft:** Wakgari Deressa, Alemayehu Worku.

**Writing – review & editing:** Wakgari Deressa, Alemayehu Worku, Workeabeba Abebe, Sefonias Getachew, Wondwossen Amogne.

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
