## [Decision Letter · Decision Letter 0]

4 Aug 2021

PONE-D-21-22047

Social distancing and preventive practices of government employees in response to COVID-19 in Ethiopia

PLOS ONE

Dear Dr. Deressa,

Thank you for submitting your manuscript to PLOS ONE. After careful consideration, we feel that it has merit but does not fully meet PLOS ONE’s publication criteria as it currently stands. Therefore, we invite you to submit a revised version of the manuscript that addresses the points raised during the review process.

We look forward to receiving your revised manuscript.

Kind regards,

Seyed Ehtesham Hasnain

Academic Editor

PLOS ONE

Journal Requirements:

2. Please include additional information regarding the survey or questionnaire used in the study and ensure that you have provided sufficient details that others could replicate the analyses. For instance, if you developed a questionnaire as part of this study and it is not under a copyright more restrictive than CC-BY, please include a copy, in both the original language and English, as Supporting Information. If the original language is written in non-Latin characters, for example Amharic, Chinese, or Korean, please use a file format that ensures these characters are visible.

3. Please state whether you validated the questionnaire prior to testing on study participants. Please provide details regarding the validation group within the methods section.

4. We noted in your submission details that a portion of your manuscript may have been presented or published elsewhere. [The socio-demographic data (Table 1) of the current MS has a partial overlap with the previously submitted related MS to the Ethiopian Journal of Health Development, which is currently under review.] Please clarify whether this [publication] was peer-reviewed and formally published. If this work was previously peer-reviewed and published, in the cover letter please provide the reason that this work does not constitute dual publication and should be included in the current manuscript.

Additional Editor Comments :

Major revision

Reviewers' comments:

Reviewer's Responses to Questions

**Comments to the Author**

1. Is the manuscript technically sound, and do the data support the conclusions?

Reviewer #1: Yes

Reviewer #2: No

Reviewer #3: Yes

2. Has the statistical analysis been performed appropriately and rigorously? 

Reviewer #1: I Don't Know

Reviewer #2: Yes

Reviewer #3: N/A

3. Have the authors made all data underlying the findings in their manuscript fully available?

Reviewer #1: Yes

Reviewer #2: Yes

Reviewer #3: No

4. Is the manuscript presented in an intelligible fashion and written in standard English?

Reviewer #1: Yes

Reviewer #2: Yes

Reviewer #3: Yes

5. Review Comments to the Author

Reviewer #1: While this is a self evaluation based study on a selected group of individuals, it has generated valuable information about public health and social measures against COVID-19. People should follow the COVID-19 appropriate behaviors including social distancing, facemasks, and hand-sanitization.

These findings have significant implications in highlighting the importance of promoting compliance with recommended protective health behaviors to effectively control the ongoing COVID-19 pandemic. Such studies can be used as a baseline data to the government, other stakeholders involved in the prevention and control of COVID-19 and researchers for other larger studies to identify factors significantly associated with preventive health measures in order to implement better intervention approaches

Reviewer #2: Comments to the authors:

The present study deals with the impact of social distancing and other preventive behaviors of government employs in response to Covid-19. A cross-sectional study covering large number of individuals from different government institutions. The study implicates the importance of social measures in preventing the spread of COVID-19 in government employees. Although the study covers the large number of individuals there is no experimental proof to data and the study is mere observatory of know preventive factors. The authors also failed to explain the effectiveness of these measures in COVID-19 positive symptomatic and asymptomatic individuals. Overall, the manuscript cannot be considered for publication in present form and will require substantial improvement for consideration. The primary concern regarding the current manuscript is the Novelty. The following queries may be considered before submitting the revision.

1. Author should provide schematic diagram of study design for more clarity.

2. Novelty is missing in the study.

3. The authors should revise the discussion part thoroughly in the manuscript which

4. seems very weak to support the study.

5. References are missing at the relevant places. Previous studies describing the importance of face masking and social distancing in COVID-19 prevention should be cited

Reviewer #3: The manuscript by Deressa et al is adequate clearly written and presented. It can be acepted follwing inclusion of minor concerns in the manuscript.

1. Many previous studies have shown community percieved risk of COVID-19. Authors should incdicate and cite these reports and explain how their analyses are different and provide novel insights from their study. For instance, https://journals.plos.org/plosone/article?id=10.1371/journal.pone.0242654#abstract0, https://www.frontiersin.org/articles/10.3389/fpsyg.2021.619145/full, https://bmcpublichealth.biomedcentral.com/articles/10.1186/s12889-021-10925-3.

2. Authors should discuss percieved risks around emerging variants of concern around globale ctiting latest reports, https://www.nature.com/articles/s41586-021-03426-1, https://www.ncbi.nlm.nih.gov/pmc/articles/PMC8000172/, https://www.nature.com/articles/s41591-021-01397-4.

6. PLOS authors have the option to publish the peer review history of their article (what does this mean?). If published, this will include your full peer review and any attached files.

Reviewer #1: No

Reviewer #2: No

Reviewer #3: No

---

## [Author Response · Author response to Decision Letter 0]

15 Aug 2021

Academic Editor’s Comments:

Comment 1. Please ensure that your manuscript meets PLOS ONE's style requirements, including those for file naming. The PLOS ONE style templates can be found at

Authors’ response: Thank you for this comment. We have addressed and fulfilled all the PLOS ONE’s style requirements. 

Comment 2. Please include additional information regarding the survey or questionnaire used in the study and ensure that you have provided sufficient details that others could replicate the analyses. For instance, if you developed a questionnaire as part of this study and it is not under a copyright more restrictive than CC-BY, please include a copy, in both the original language and English, as Supporting Information. If the original language is written in non-Latin characters, for example Amharic, Chinese, or Korean, please use a file format that ensures these characters are visible.

Authors’ response: Thank you very much for this comment. We have included additional information with regard to the questionnaire (English, Amharic and Afan Oromo) as Supporting Information as follows:

S2 Appendix. English version of self-administered questionnaire developed for the survey, June 2020.

S3 Appendix. Amharic version of self-administered questionnaire used for the survey, June 2020.

S4 Appendix. Afan Oromo version of self-administered questionnaire used for the survey, June 2020.

Comment 3. Please state whether you validated the questionnaire prior to testing on study participants. Please provide details regarding the validation group within the methods section.

Authors’ response: Thank you for this comment. We have revised this section in the “Instrument and data collection” part of the “Methods” section as follows:

“A total of 20 trained data collectors with master’s degree and previous experience were involved in data collection with 2-3 institutions per each data collector. Training and orientation on the survey including how to administer the questionnaires were conducted for the data collectors on 2nd June 2020. In addition, the Amharic and Afan Oromo versions of the questionnaires were tested on one target person by each data collector prior to the actual data collection. Few minor revisions of the instruments were made”. Lines 250-256 (Revised MS with Track Changes)

Comment 4. We noted in your submission details that a portion of your manuscript may have been presented or published elsewhere. [The socio-demographic data (Table 1) of the current MS has a partial overlap with the previously submitted related MS to the Ethiopian Journal of Health Development, which is currently under review]. Please clarify whether this [publication] was peer-reviewed and formally published. If this work was previously peer-reviewed and published, in the cover letter please provide the reason that this work does not constitute dual publication and should be included in the current manuscript.

Authors’ response: Thank you very much for this comment. 

The socio-demographic data (Table 1) of the current MS has a partial overlap with the previously submitted related MS to the Ethiopian Journal of Health Development [Knowledge and perceptions of COVID-19 among government employees in Ethiopia], which is currently under review. This MS has only knowledge and perception components and it was initially drafted taking into consideration the time sensitiveness of the findings. Most of the questions were on symptoms and causes of COVID-19, incubation period, mode of transmission, population group at risk and susceptible to infection, severity or death. While we were developing this MS, the findings have become outdated and that is why we submitted to the local Journal, which is very slow. Recently we received reviewer’s comments (one reviewer positively considered and the other reviewer rejected the MS). However, the Journal is positive to consider for publication although the findings are outdated, and it is currently under review after the submission of the revised version. This means it has not been published. 

One of the comments of the reviewers is presented below just for your information:

• Reviewer’s comment: This information looks outdated/decayed/not fresh based on the time of the study and the pandemic nature. i.e it will have less novelty at this stage. Or the researchers should explain the importance of publishing this evidence at this time.

• And we responded to the above comments as follows: 

o Authors’ response: This study was conducted in June 2020, three months after the first confirmed COVID-19 case was reported in Ethiopia. The MS was submitted to EJHD in Feb 2021, nine months after the data were collected. It seems the data looks old as far as the COVID-19 is concerned with regard to the knowledge and perceptions of the public. However, the findings are important and relevant with regard to the COVID-19 perspective at the time of the study. It will also serve as a baseline for comparison with future studies, and shows how the people reacted at the beginning of the pandemic. 

Comment 5. Please include captions for your Supporting Information files at the end of your manuscript, and update any in-text citations to match accordingly. Please see our Supporting Information guidelines for more information: http://journals.plos.org/plosone/s/supporting-information. 

Authors’ response: Thank you so much. We have included the captions for all files in the Supporting Information, and presented them at the end of the Revised Manuscript. We have also checked and updated the in-text citations accordingly.

Additional Editor Comments:

Major revision

Reviewers' comments:

Reviewer's Responses to Questions

Reviewers' comments:

Reviewer #1: Comments to the authors

While this is a self-evaluation-based study on a selected group of individuals, it has generated valuable information about public health and social measures against COVID-19. People should follow the COVID-19 appropriate behaviors including social distancing, facemasks, and hand-sanitization.

These findings have significant implications in highlighting the importance of promoting compliance with recommended protective health behaviors to effectively control the ongoing COVID-19 pandemic. Such studies can be used as a baseline data to the government, other stakeholders involved in the prevention and control of COVID-19 and researchers for other larger studies to identify factors significantly associated with preventive health measures in order to implement better intervention approaches.

Authors’ response: Thank you so much for your valuable and encouraging comments. We highly appreciate your time for reading and reviewing our manuscript. 

Reviewer #2: Comments to the authors

The present study deals with the impact of social distancing and other preventive behaviors of government employees in response to Covid-19. A cross-sectional study covering large number of individuals from different government institutions. The study implicates the importance of social measures in preventing the spread of COVID-19 in government employees. Although the study covers the large number of individuals there is no experimental proof to data and the study is mere observatory of know preventive factors. The authors also failed to explain the effectiveness of these measures in COVID-19 positive symptomatic and asymptomatic individuals. Overall, the manuscript cannot be considered for publication in present form and will require substantial improvement for consideration. The primary concern regarding the current manuscript is the Novelty. 

The following queries may be considered before submitting the revision.

Comment 1. Author should provide schematic diagram of study design for more clarity.

Authors’ response: Thank you for this comment, and we have presented the schematic diagram of the sampling design and sample size distribution of the study in the Supporting Information as follows:

S1 Fig. Schematic design of the sampling methods and sample distribution, June 2020

Comment 2. Novelty is missing in the study.

Authors’ response: Thank you for this comment. Yes, we agree that unfortunately there is less novelty in our study. However, the study has generated valuable information with regard to public health and social measures against COVID-19 in the study population. The findings have significant implications in highlighting the importance of promoting compliance with recommended protective health behaviors to effectively control the ongoing COVID-19 pandemic. In addition, the findings can be used as a baseline data to the government and other stakeholders involved in the prevention and control of COVID-19 and researchers for other larger studies to identify factors significantly associated with preventive health measures in order to implement better intervention approaches.

Comment 3. The authors should revise the discussion part thoroughly in the manuscript which seems very weak to support the study.

Authors’ response: Thanks a lot for this comment. Based on this comment, we tried to revise the Discussion part, and also added the following new paragraphs:

“This study revealed higher levels of social and preventive practices in response to COVID-19. One potential reason for this higher degree of practices in the current study was due to higher level of perceived risk of COVID-19 associated with widespread information about the pandemic provided to the public. A number of studies reported higher levels of perceived risk towards COVID-19 associated with increased preventive practices, such as in Turkey [43 Yildirim et al., 2021], Hong Kong [44 Kwok et al., 2020] and the US [45 Bruine & Bennett 2020]. In contrast, a study conducted towards the end of April 2020 in northwest Ethiopia revealed a lower level (23%) of perceived risk about COVID-19 [46 Kabito et al., 2020], which might be explained by the differences in measurements, study population or low levels of risk communication activities in the area. The finding of the current study is encouraging and may highlight the effectiveness of risk communication interventions extensively implemented in Addis Ababa during the early phase of the pandemic”. Lines 482-492 (Revised MS with Track Changes)

43.Yildirim M, Gecer E, Akgul O. The impacts of vulnerability, perceived risk, and fear on preventive behaviors against COVID-19. Psychology, Health & Medicine 2021; 26(1):35-0043. DOI: 10.1080/13548506.2020.1776891.

44.Kwok KO, Li KK, Chan HHH, Yi YY, Tang A, Wei WI, et al. Community Responses during Early Phase of COVID-19 Epidemic, Hong Kong. Emerg Infect Dis. 2020;26(7):1575-1579. doi: 10.3201/eid2607.200500.

45.Bruine de Bruin W, Bennett D. Relationships Between Initial COVID-19 Risk Perceptions and Protective Health Behaviors: A National Survey. Am J Prev Med. 2020;59(2):157-167. doi: 10.1016/j.amepre.2020.05.001.

46.Kabito GG, Alemayehu M, Mekonnen TH, Daba Wami S, Azanaw J, Adane T, et al. Community's perceived high risk of coronavirus infections during early phase of epidemics are significantly influenced by socio-demographic background, in Gondar City, Northwest Ethiopia: A cross-sectional -study. PLoS One 2020;15(11):e0242654. doi: 10.1371/journal.pone.0242654.

“…….. Although the current vaccines are effective against the consequences of COVID-19, most low-income and middle-income countries face difficulties in accessing vaccines to their populations [60 Lancet Commission 2021]. Thus, the intensive implementation of the proven social and preventives practices such as wearing facemask, handwashing, maintaining physical distancing, and avoiding mass gatherings and crowded places is still very critical. In addition, many new variants of SARS-CoV-2 with increased transmissibility and disease severity have been recently emerged in many countries around the world, which can lead to significant clinical, therapeutic and public health impacts [61 Janik et al., 2021; 62 Singh et al., 2021]. Lines 620-627 (Revised MS with Track Changes)

60.Lancet Commission on COVID-19 Vaccines and Therapeutics Task Force Members. Urgent needs of low-income and middle-income countries for COVID-19 vaccines and therapeutics. Lancet 2021;397(10274):562-564. doi: 10.1016/S0140-6736(21)00242-7.

61.Janik E, Niemcewicz M, Podogrocki M, Majsterek I, Bijak M. The emerging concern and interest SARS-CoV-2 variants. Pathogens 2021; 10(6):633. doi: 10.3390/pathogens10060633.

62.Singh J, Samal J, Kumar V, Sharma J, Agrawal U, Ehtesham NZ, et al. Structure-Function Analyses of New SARS-CoV-2 Variants B.1.1.7, B.1.351 and B.1.1.28.1: Clinical, Diagnostic, Therapeutic and Public Health Implications. Viruses. 2021;13(3):439. doi: 10.3390/v13030439.

Comment 4. References are missing at the relevant places. Previous studies describing the importance of face masking and social distancing in COVID-19 prevention should be cited.

Authors’ response: Thank you for this comment. Based on this comment, we tried to look at the MS and checked for the references particularly related to the importance of face masking and social distancing. In the DISCUSSION section, we already cited sufficient references about the importance of face masking and social distancing. But we added more references and modified the sentences particularly with regard to social distancing:

Lines 515-523 (Revised MS with Track Changes)

“Studies from China [21] and the US [49] have shown wearing masks is effective in reducing the risk of infection and mitigating the spread of COVID-19, particularly when combined with other preventive measures such as physical distancing and frequent handwashing. A recent systematic review and meta-analysis study funded by the WHO demonstrated the effectiveness of physical distancing of 1m or more and the use of facemasks in public and health-care settings in the prevention of coronavirus transmission, where both interventions reduced the risk of infection of coronavirus by more than 80% [50]. A case-control study from Thailand found that mask wearing, frequent handwashing and social distancing of ≥1m were independently and significantly associated with reduced risk of SARS-CoV-2 infection among the general public [51]”.

Lines 524-530 (Revised MS with Track Changes)

“Previous studies also showed that the use of facemasks among the general population significantly reduced total infections and the number of deaths, and mask wearing is considered as one of the most effective public health measures in mitigating transmission of SARS-CoV-2 [52]. Studies from various countries showed the effectiveness of the national application of social distancing measures in reducing the spread of SARS-CoV-2 [53-55]. These studies provide the most reliable evidence on the effectiveness of the use of social distancing measures at community level to mitigate the spread of COVID-19. However, there are still inconsistent scientific evidence about the effectiveness of using facemasks by healthy people in the community to prevent infection with SARS-CoV-2. A randomized controlled trial conducted in Denmark reported no statistically significant difference in the incidence of SARS-CoV-2 between facemask wearers and non-wearers [53]”.

53.Vo HL, Nguyen HAS, Nguyen KN, Nguyen HLT, Nguyen HT, Nguyen LH, et al. Adherence to Social Distancing Measures for Controlling COVID-19 Pandemic: Successful Lesson From Vietnam. Front Public Health 2020;8:589900. DOI: 10.3389/fpubh.2020.589900.

54.Kim MC, Kweon OJ, Lim YK, Choi SH, Chung JW, Lee MK. Impact of social distancing on the spread of common respiratory viruses during the coronavirus disease outbreak. PLoS One 2021;16(6):e0252963. DOI: 10.1371/journal.pone.0252963.

55.Thu TPB, Ngoc PNH, Hai NM, Tuan LA. Effect of the social distancing measures on the spread of COVID-19 in 10 highly infected countries. Sci Total Environ 2020;742:140430. DOI: 10.1016/j.scitotenv.2020.140430.

We also found a paragraph in the DISCUSSION part with missing references and cited proper references in the revised version of the MS as follows:

Lines 559-566 (Revised MS with Track Changes)

“Behavioral changes are currently one of the main tools to fight against COVID-19 [18-21]. These changes include practicing social and physical distancing, frequent handwashing, using hand sanitizers, wearing facemasks and testing for COVID-19 [15-18]. Nonetheless, these behavioral measures are effective if they are widely accepted and applied by the public. To have these measures widely understood and implemented by the community, the government needs to heighten risk communication activities to educate the public about the significance of frequent handwashing, wearing facemasks, and social distancing in containing the transmission of SARS-COV-2”.

With regard to COVID-19 testing, we have added the following sentence in the Discussion part and added one more reference:

“….. Testing can help people determine if they are infected with SARS-CoV-2 regardless of whether they have symptoms and particularly to self-isolate themselves at the time of most infectious period to minimize the risk of spreading the infection to others and to inform public health decisions [56 Peeling et al., 2021]. Lines 554-557 (Revised MS with Track Changes)

56.Peeling RW, Olliaro PL, Boeras DI, Fongwen N. Scaling up COVID-19 rapid antigen tests: promises and challenges. Lancet Infect Dis. 2021:S1473-3099(21)00048-7. doi: 10.1016/S1473-3099(21)00048-7.

Reviewer #3: The manuscript by Deressa et al is adequate, clearly written and presented. It can be accepted following inclusion of minor concerns in the manuscript.

Comment 1. Many previous studies have shown community perceived risk of COVID-19. Authors should indicate and cite these reports and explain how their analyses are different and provide novel insights from their study. For instance, https://journals.plos.org/plosone/article?id=10.1371/journal.pone.0242654#abstract0, https://www.frontiersin.org/articles/10.3389/fpsyg.2021.619145/full, https://bmcpublichealth.biomedcentral.com/articles/10.1186/s12889-021-10925-3.

Authors’ response: Thank you for this comment. We have added a new paragraph about the relationship between perceived risk and preventive practices, and cited ---- more references and expanded the discussion part of the MS as follows:

“This study revealed higher levels of social and preventive practices in response to COVID-19. One potential reason for this higher degree of practices in the current study was due to higher level of perceived risk of COVID-19 associated with widespread information about the pandemic provided to the public. A number of studies reported higher levels of perceived risk towards COVID-19 associated with increased preventive practices, such as in Turkey [43 Yildirim et al., 2021], Hong Kong [44 Kwok et al., 2020] and the US [45 Bruine & Bennett 2020]. In contrast, a study conducted towards the end of April 2020 in northwest Ethiopia revealed a lower level (23%) of perceived risk about COVID-19 [46 Kabito et al., 2020], which might be explained by the differences in measurements, study population or low levels of risk communication activities in the area. The finding of the current study is encouraging and may highlight the effectiveness of risk communication interventions extensively implemented in Addis Ababa during the early phase of the pandemic”. Lines 482-492 Revised MS with Track Changes)

43.Yildirim M, Gecer E, Akgul O. The impacts of vulnerability, perceived risk, and fear on preventive behaviors against COVID-19. Psychology, Health & Medicine 2021; 26(1):35-0043. DOI: 10.1080/13548506.2020.1776891.

44.Kwok KO, Li KK, Chan HHH, Yi YY, Tang A, Wei WI, et al. Community Responses during Early Phase of COVID-19 Epidemic, Hong Kong. Emerg Infect Dis. 2020;26(7):1575-1579. doi: 10.3201/eid2607.200500.

45.Bruine de Bruin W, Bennett D. Relationships Between Initial COVID-19 Risk Perceptions and Protective Health Behaviors: A National Survey. Am J Prev Med. 2020;59(2):157-167. doi: 10.1016/j.amepre.2020.05.001.

46.Kabito GG, Alemayehu M, Mekonnen TH, Daba Wami S, Azanaw J, Adane T, et al. Community's perceived high risk of coronavirus infections during early phase of epidemics are significantly influenced by socio-demographic background, in Gondar City, Northwest Ethiopia: A cross-sectional -study. PLoS One 2020;15(11):e0242654. doi: 10.1371/journal.pone.0242654.

Comment 2. Authors should discuss perceived risks around emerging variants of concern around global citing latest reports, https://www.nature.com/articles/s41586-021-03426-1, https://www.ncbi.nlm.nih.gov/pmc/articles/PMC8000172/, https://www.nature.com/articles/s41591-021-01397-4.

Authors’ response: Thank you for this comment, and we have now included the following statement in the Discussion part and added three additional references. 

“…….. Although the current vaccines are effective against the consequences of COVID-19, most low-income and middle-income countries face difficulties in accessing vaccines to their populations [62 Lancet Commission 2021]. Thus, the intensive implementation of the proven social and preventives practices such as wearing facemask, handwashing, maintaining physical distancing, and avoiding mass gatherings and crowded places is still very critical. In addition, many new variants of SARS-CoV-2 with increased transmissibility and disease severity have been recently emerged in many countries around the world, which can lead to significant clinical, therapeutic and public health impacts [63 Janik et al., 2021; 64 Singh et al., 2021; 65 Davies et al., 2021]. Lines 621-628 (Revised MS with Track Changes)

62.Lancet Commission on COVID-19 Vaccines and Therapeutics Task Force Members. Urgent needs of low-income and middle-income countries for COVID-19 vaccines and therapeutics. Lancet 2021;397(10274):562-564. DOI: 10.1016/S0140-6736(21)00242-7.

63.Janik E, Niemcewicz M, Podogrocki M, Majsterek I, Bijak M. The emerging concern and interest SARS-CoV-2 variants. Pathogens 2021; 10(6):633. DOI: 10.3390/pathogens10060633.

64.Singh J, Samal J, Kumar V, Sharma J, Agrawal U, Ehtesham NZ, et al. Structure-Function Analyses of New SARS-CoV-2 Variants B.1.1.7, B.1.351 and B.1.1.28.1: Clinical, Diagnostic, Therapeutic and Public Health Implications. Viruses 2021;13(3):439. DOI: 10.3390/v13030439.

65.Davies NG, Jarvis CI, CMMID COVID-19 Working Group, Edmunds WJ, Jewell NP, Diaz-Ordaz K, et al. Increased mortality in community-tested cases of SARS-CoV-2 lineage B.1.1.7. Nature 2021;593(7858):270-274. DOI: 10.1038/s41586-021-03426-1.

---

## [Editor Report · Decision Letter 1]

24 Aug 2021

Social distancing and preventive practices of government employees in response to COVID-19 in Ethiopia

PONE-D-21-22047R1

Dear Dr. Deressa,

We’re pleased to inform you that your manuscript has been judged scientifically suitable for publication and will be formally accepted for publication once it meets all outstanding technical requirements.

Kind regards,

Seyed Ehtesham Hasnain

Academic Editor

PLOS ONE

Additional Editor Comments (optional):

I have gone through the revised manuscript and also the Authors response to the comments of the Reviewers. In my view, the authors have comprehensively revised the manuscript addressing all the comments of the reviewers. All the explanations provided by the Authors to the queries of reviewers are quite satisfactory. Authors have presented the schematic diagram of the sampling design and sample size distribution of the study in the Supporting Information. Discussion part of the manuscript has been revised by the Authors. They have added more references in the study and modified the sentences particularly with regard to social distancing. I recommend this manuscript for publication.
---

## [Editor Report · Acceptance letter]

26 Aug 2021

PONE-D-21-22047R1 

Social distancing and preventive practices of government employees in response to COVID-19 in Ethiopia 

Dear Dr. Deressa:

I'm pleased to inform you that your manuscript has been deemed suitable for publication in PLOS ONE. Congratulations! Your manuscript is now with our production department. 

Kind regards, 

on behalf of

Prof. Seyed Ehtesham Hasnain 

Academic Editor

PLOS ONE